# Study of the Suitability of Different Types of Slag and Its Influence on the Quality of Green Grouts Obtained by Partial Replacement of Cement

**DOI:** 10.3390/ma12071166

**Published:** 2019-04-10

**Authors:** Francisca Perez-Garcia, Maria Eugenia Parron-Rubio, Jose Manuel Garcia-Manrique, Maria Dolores Rubio-Cintas

**Affiliations:** 1Departamento de Ingeniería Civil, Materiales y Fabricación, Universidad de Málaga, 29071 Málaga, Spain; perez@uma.es (F.P.-G.); josegmo@uma.es (J.M.G.-M.); 2Departamento de Ingeniería Industrial y Civil, Universidad de Cádiz, 11202 Algeciras, Spain; mariadolores.rubio@uca.es

**Keywords:** cementitious grout, green grout, cement, slag substitution, valorization, circular economy

## Abstract

This paper is part of a research line focused on the reduction of the use of cement in the industry. In this work, the study of work methodologies for the manufacture of green cementitious grout mixtures is studied. Grout is widely used in construction and it requires an important use of raw materials. On the other hand, the steel industry faces the problem of the growing generation of slag wastes due to the increase in steel manufacturing. The green grout aims to achieve the dual objective of reducing the demand for cement and improve the slag waste valorization. Slag is not introduced as an aggregate but through the direct replacement of cement and no additives. The research seeks a product where we can use steel slag intensively, guaranteeing minimum resistance and workability. Results with substitutions between a 25% to 50% and water/cement ratio of 1 are presented. In particular, the suitability of different slags (two Ladle Furnace Slag (LFS) and one Blast Furnace Slag (GGBS)) in the quality of the final product are analyzed. The feasibility of replacing cement with slag and the importance of the origin and pretreatment are highlighted.

## 1. Introduction

One of the most important challenges that society faces is to achieve a balance between the consumption of raw materials and our need for development. The future evolution of the industrial activity must include criteria of both efficiency and reuse of waste. The dimension of transformation must be limited to a sustainable environment where the global needs could be satisfied without severely compromising those of future generations (Brundlandt Report, 1987) [1]. To achieve this sustainable environment, actions must be taken in the reduction of harmful gas emissions (greenhouse effect) and the reduction of the use of natural raw materials.

One way to reduce the use of raw materials is to lead our efforts toward objectives such as those proposed in the circular economy theory, being aware that we inhabit a world with finite resources.

Therefore, the essence of the circular economy is to optimize the reuse of the generated resources and introduce them back into the production chain. This process, known as waste valorization [2], has become more and more important and today is a field of research with great potential.

Our efforts are focused on the construction industry, in particular, in the process of manufacturing cement-based materials. It is a fundamental element due to its role as a binding component in different mixtures. In order to reach a sustainable environment, the production of cement should be reduced. One of the reasons is to minimize the extraction of limestone, but more significant is the reduction of energy consumption and on greenhouse gas emissions.

Throughout the entire process, the estimated CO_2_ release during clinker manufacture is around 0.7 to 0.9 tons per Portland cement ton, which means that the cement industry generates between 7 and 9% of CO_2_ worldwide. The decrease of these emissions is the trigger of the notable research interest in making progress toward reducing the industry’s dependence on cement [3,4,5,6]. As an alternative to the use of cement, its partial replacement by other materials is proposed. At this point, the idea of valorization of existing waste in the industry itself becomes important.

Steel slag is a byproduct of steel manufacturing, which is obtained by the chemical reactions that take place in the processes of metal formation. It is a complex solution of silicates and oxides produced during the separation of the molten steel from impurities. The properties of the slag produced during steel making depends on many factors, mainly the manufacturing process. According to Setie et al. [2], four types of steel slag can be distinguished: electric arc furnace (EAF) slag, blast furnace slag (GGBS), basic oxygen furnace slag (BOFS), and ladle furnace slag (LFS) [7,8,9].

The EAF is a strong, dense, nonporous aggregate that is cubical in shape, has good resistance to polishing, and has an excellent affinity to bitumen. Therefore, EAF are more suitable for engineering purposes. The EAF slags can be also divided in two types: oxidizing or black (EAFS) and reducing or white (LFS). In a usual manufacturing process of steel, the EAFS produced is in the order of 110–130 kg per metric ton, and the LFS white one is about 20–30 kg per metric ton.

The increase in steel consumption supposes a proportional increase in the generation of this slag’s waste. The strategies of waste valorization of the slags are diverse but not sufficient to achieve in practice a real reuse of these. Either by regulatory or economic problems. The steel slags are used in many areas, from fertilizers to civil industry. In the European Union, it has been used as an additive to make up the cements. The slag properties make them very appropriate for aggressive environments. Slags improve their resistance against salt water and sulphates (maritime facilities). In the last few decades, research efforts have been focused on its use as additives or as aggregate substitutes with arid, both fine and coarse, or as a substitute of arid as a bituminous binder in the pavement layer [10,11,12,13,14,15,16,17,18,19,20].

In the literature, we can find some recent and interesting works about cementitious grouts containing supplementary materials. In 2015, Celik et al. [21] investigate the mixture of rice husk ash in cement-based grout, the rheological properties of the mixture result in the increase in apparent viscosity. Amahjour et al. [22] (2002) or Pastor et al [23] (2016) add fly ash and silica fume to increase mechanical strength.

There are also studies where the substitutions of blast furnace slag are made in small percentages but always with chemical additives. Azadi et al. [24] (2013) worked with chemical additives to optimize the grout. They used sodium silicate (Na_2_SiO_3_) to increase resistance, sodium carbonate (Na_2_CO_3_) to reduce bleeding, or triethanolamine (TEA) to promote injection. In 2017, Zhang [25] introduced sodium silicate for quick adjustment.

An interesting research in this area is that done by Krishnamoorthy et al. [26]. In this study a cementitious grout containing supplementary cementitious materials (SCM) (fly ash, silica fume, GGBS) are presented. The results on flow characteristics, strength, and durability show that cementitious grouts containing SCM can be used successfully to repair concrete structures. They reach percentage substitution of 20 to 40% of GGBS slags with water/cement ratios of 0.25 to 0.40.

Huang [27] presented the study on cement ash slurries containing polypropylene (PP), fiber, and super plasticizer. With the addition of PP fiber, better resistance against cracking, sulphate attack, and volume changes was observed, but resulted in a higher viscosity and permeability. Bastien et al. [28] studied the properties of cement slurries with a low ratio of cement and water with superplasticizer and a low proportion of precipitated silica (3% by weight of cement). The rheological properties were also investigated. It is possible to obtain slurry mixtures with zero bleeding, good fluidity, and high compressive strength that meet the requirements for the use of post-tensioning.

Shannag [29] studied cementitious grouts, adding silica fume and natural pozzolan to achieve high performance. This incorporation results in a high fluidity, zero bleeding, high strength, and satisfactory shrinkage.

The literature indicates that the desirable properties for the grouts are that they must possess good fluidity, reduction of bleeding, initial setting time that is not too short, adequate strength, and durability.

Our line of research seeks to advance in the study of the technical feasibility of replacing cement with slag from the steel industry [30], both for the production of green concrete [7,31] and green cementitious grout.

However, there are many factors involved, as the properties of the resulting product related not only mechanical properties, but also durability, that are correlated with the characteristics of the slag used and its proportion.

This paper presents the results obtained in a set of tests for the manufacture of green grouts in a substitution dosage that varies between 0 and 50%. The substitutions have been made on cement Portland (CEM I). Unlike Portland CEM III and IV cements, which already have slag incorporations in their manufacture [32,33], our objective is to evaluate the possibility of working the cementitious grout with recycled material directly [34]. In particular, each mixture has been subjected to a test campaign of a slump test, compressive and bending tests, and exudation test. As mentioned, it is part of a wide “program intended to elaborate a standard to guide the use of steel slag as cement substitute” [7,31].

In the resulting discussion, the optimized simplification of the parameters of the model is considered when admitting that each phase of the material is subject to a similar tension that eliminates the influence of the form, the size, and the disposition of the phases. Therefore, the only factors that this approach considers are the concrete’s resistance properties in relation to the replacement of some of its components by slag. The models are complicated, taking into account the case that the cement paste is the connection phase, where the inclusions form a disconnected phase [35].

A fundamental parameter to analyze will be the different behavior obtained (quality of the Green Grout) according to the characteristics of the slag. Two key aspects, the origin of the slag (dependent on the type of steel produced) and the treatments to which the slag have been subjected becomes determinant. In this research, slag from different origins within the country have been studied. The results obtained, such as those presented in a previous work for the case of concrete [7], are very positive and show the feasibility of the mixture.

On the following section, the materials analyzed are described. Then, a brief overview of the tests performed is made. In the next section, results are presented and discussed. Finally, the fundamental conclusion and lines of future research are summarized.

## 2. Materials

This article focuses on the progress made in replacing material in cementitious grout mixtures. This material has a wide presence in civil constructions. Two of the applications where this research presents potential applicability are the jet grouting, for the improvement of soil, and the execution of deep foundations such as micropiles. In the jet grouting process, the slurry is injected into the soil pores in order to fill and create cohesion, which increases the resistance characteristics or, equivalently, improves its mechanical properties [24,36]. Therefore, the rheological properties of grouts are directly related to the pumping capacity to penetrate holes and cracks [21].

One of the fundamental parameters in cement-based grout is the water/cement ratio by weight. Depending on the application, different relationships are recommended. In applications of foundation by injection, the cementitious grout needs to behave like a fluid able to penetrate the soil or rock so the ratio will vary between 0.6 and 2. In sealed works, it varies between 0.5 and 1. Figure 1 presents a schematic of these recommendations.

Ten different mixes were designed by substituting a 30%, 40% and 50% of cement by slags (% in weight) obtained from three different blast furnaces in Spain, according to previous works with concrete [7,31]. Table 1 shows the cement and slag chemical composition (data provided by the supplying company). Table 2 presents the nomenclature used for the mixtures.

The used cement was Portland Cement CEM I 42.5 R (EN 197-1 [37], Holcim, Málaga, Spain). This cement was selected due to the absence of any kind of additive that could mask the results. It was used as reference pattern. Density: 3150 kg/m^3^. Specific surface area: >280 m^2^/kg. The water–cement ratio used was 1.

Three different slags are used (S1, S2 and S3). The first of them, S1, was a ground granulated blast furnace slag (GGBS) with mechanical processing. It presented a maximum grain size of 0.063 mm, density of 2910 kg/m^3^, and specific surface area of 462 m^2^/kg. On the other hand, S2 and S3 types were unprocessed ladle furnace slag (LFS) from two different steel mills. The only process they were subjected to was sieving in the lab with a 0.063 mm sieve. The fraction obtained through sieving was 23% and 15% by weight, respectively.

Chemical composition of the slags and cement used in the grouts mixtures is shown in Table 1. This chemical composition was proportionated by the slag supplier companies and it refers to the slags before the sieving process when needed in the laboratory.

As can be seen in Table 1, the chemical composition of the different slags varied for each of them, especially the percentage of SiO_2_ and CaO.

The materials employed in this work did not include any type of additive because the objective of the research was to determine how the different slags behaved as substitutes of the binder without being affected by any additional parameter.

## 3. Tests Description

The mixes described in previous section were subject to different standard tests. The objective of these tests was to evaluate how cement-slag substitution may affect the main properties such as consistency, workability, and mechanical capabilities (compressive and flexural strength).

The batching was prepared according to the European standard used in the manufacture of cement grouts: EN 447 [38]. The laboratory conditions were 24 and 26 °C (temperature) and 40% relative humidity. An electric mixer of robust construction was used, with two speeds of rotation, with a mixing paddle with an anchoring system according to EN 196/1 [39].

Using the dispensing hopper, the Portland cement was incorporated into the mixing bowl with the slag where it was mixed for 90 s, then the water was added, again mixing for 180 s (Figure 2). For each of the mixtures, nine prismatic test specimens of 4 × 4 × 16 cm^3^ were prepared. Each one according to the EN 12390-2 standard [40] for hardened concrete where the methods for the manufacture and curing of specimens destined for the performance of resistance tests are described.

### 3.1. Flow Cone Test

The test for the flow of grout mixtures (flow cone method) was determined according to the norm EN 445 [41]. The test determines the time of efflux of a specified volume of fluid cement grout through a standardized flow cone.

Before beginning, the inside of the cone was moistened by filling the cone with water. The water was drained from the cone one minute before the test. The grout was poured slowly to prevent trapped air. The quantity tested was one liter of mix. Once full, the stopwatch was started, and simultaneously, the stopper was removed. The recorded result was be the time in which the entire grout passed through the cone.

### 3.2. Flexural Strength Test

The test was carried out at 7, 28 and 90 days for all the test pieces (nine prismatic test specimens for each mixture) according to EN 196-1 [39] and EN 196-7 [42] standards for cements. The test machine was equipped with a bending device incorporating two steel support rollers and a third steel loading roller of the same diameter and equidistant from the other two. The length of these rollers was between 45 and 50 mm. The load was applied continuously and without sudden shocks. The force did not begin to be applied until the load roller and the support rollers rest firmly on the specimen. The rate increase *R* was 16 *N*/*s*, according to the expression:(1)R=23×S·d1·d22 l×Ns
where *d*_1_ and *d*_2_ are the dimensions of the square section of the specimen and *l* = 3 × *d* is the distance between the rollers in millimeters.

The force signal was provided by an adjustable load cell to the upper, lower, or base bridge. It was formed by a strain gauge Wheatstone bridge, adhered to a structure. The force captured by the load cell was, due to its location, the same as that of the specimen under test; that is, there was a direct coupling between the test piece and the load cell. The machine was controlled by a computer through an ETIWIN control software, with ENAC (National Accreditation Entity in Spain) calibration certificate.

### 3.3. Compressive Strength Test

The test was performed at 7, 28, and 90 days as a bending test according to standards EN 196-1 [39] and EN 196-7 [42]. The number of specimens tested was 18 and an average value was calculated. According to Neville [43], the compressive strength of the modified cubic specimen would be 5% higher than the standard cubic specimen. An average value was obtained from this study because only two specimens per flexural strength test were performed.

The test was carried out with the same machine, model ETIMATIC-Proetisa H0224 (Production of Technical and Industrial Equipment, Madrid, Spain). The applied pressure was at an invariant rate of 0.5 MPa/s. The compressive strength is given by the expression:(2)$$f_{c}=\frac{F}{A_{c}}$$
where *f_c_* is the compressive strength in MPa, *F* is the maximum breaking load expressed in N, and *A_c_* is the cross-sectional area of the specimen given in mm^2^.

### 3.4. Exudation Test

This test gives the exudation of the grout. It was carried out according to EN 445 [41] (Figure 2). Exudation was measured as the volume of water remaining on the surface of the mix that was kept protected from evaporation. The variation in volume was measured as a difference in percentage of the volume of the grout between the start and the end of the test. The test mainly measured the volume variation caused by sedimentation or expansion. A transparent tube, approximately 60 mm in internal diameter and around 1 m in length, was used. The tube was placed in a vertical position with the top end open. It ensured a rigid fixation that prevented any movement or vibration. The grout was poured into the tube with a constant flow to ensure that no trapped air remained. The tube was filled to a height, h_o_. The ambient temperature of the laboratory was 18.1 °C and the grout acquired a temperature of 18.3 °C. The start time t_0_ and the height h_0_ were recorded. The height of the cement grout, h_g_, was recorded at intervals of 15 min during the first hour, and then at 2 h, 3 h, and 4 h. The height of the exuded water, h_w_, was recorded at the same time as the measurements of the grout were made. Possible heterogeneities that could be seen in its appearance through the transparent tube were recorded. The volume variation was:h_w_/h_o_ × 100% (3)

## 4. Interfacial Transition Zone Review

Before presenting the data, the basic principles of mixing models based on the interfacial transition zone are reviewed (ITZ) [44]. This model usually applied to concrete assumes that the material is idealized as a composite of mortar and aggregate, where any arbitrarily small volume contains both mortar and aggregate in fixed proportions [45]. The same hypothesis can be applied to the cementitious grout, which properties will be very influenced by the microstructure. This can be classified into three phases: aggregate, cement paste, and the interfacial transition zone (ITZ). ITZ has a critical role. This transition zone has a size comparable with the size of cement grains.

The effects of varying the percentage of slag substitution affects the state of the structure and causes an improvement in fluency. This leads us to expect an increase in the mechanical strength of the hardened grout. There are few studies on this phenomenon, due to it being difficult to find suitable definitions. It is a new diffuse distribution of the particles in the grout. The global grain size does not change, but the permeability evolves. If locally transported particles do not migrate further, an obstruction occurs that accompanies an overpressure. In short, the substitution of cement by slag results in a redistribution of the fine particles without modification of the total solid volume of the specimen.

Below, the formulation of these models are summarized. The partial densities for the three constituents are given by the following expressions:(4)$$\rho^{s}=\rho_{s}(1-\varphi )$$
(5)$$\rho^{w}=\rho_{w}\varphi (1-c)$$
(6)$$\rho^{c}=\rho_{c}\varphi \;c$$

Where ρ_s_, ρ_w_, and ρ_c_ are the real densities of the skeleton, water, and fluidized cement paste, respectively. The porosity, φ, defines the proportion of the holes in relation to the total volume. “c” is the concentration in the cement paste of the ITZ zone. It represents the total volume of the particles transported from the skeleton by the filtering forces in the void volume. The fraction of the mass of the fluidized solid is:(7)$$c_{m}=c\rho_{w}/\left(c\rho_{w}+(1-c)\rho_{c}\right)$$

Applying the mass conservation equations to each phase, Equation (8) presents the variation of mass for solid phase, Equation (9) for liquid phase, and Equation (10) for fluidized solid:(8)$$\frac{\partial \rho^{s}}{\partial t}+\nabla .(\rho^{s}v^{s})=m^{s}$$
(9)$$\frac{\partial \rho^{w}}{\partial t}+\nabla .(\rho^{w}v^{w})=0$$
(10)$$\frac{\partial \rho^{c}}{\partial t}+\nabla .(\rho^{c}v^{c})=m^{c}$$
where “m” represents the typical variation of the mass of that constituent (m^w^ = 0), “t” is the time and ∂ is the gradient operator. Assuming that all the particles transferred from the skeleton re-enter the fluid, the following mass can be established:(11)$$m^{c}+m^{s}=0$$

It will be assumed, on the other hand, that the slag transported via filtration in the slurry moves with the same speed as the particles near the ITZ zone. This relationship translates the particular nature of the phenomenon that is considered in the framework of this study. There is, for example, no chemical reactions that cause a divergence between the mass transported from the solid skeleton per unit of time and that which is transformed into cement paste at the same time. The previous hypothesis also assumes that the ITZ does not significantly modify the proportions between the different phases that constitute the initial sample. Therefore, the initial variation and the temporal evolution of ρ_s_ (density of the solid skeleton) remain negligible. This means, in particular, that this phenomenon does not develop considerably and causes little variation in the initial properties of the fluidized cement paste in, for example, its density.

## 5. Results and Discussion

In this section, the main results obtained are presented and discussed. The results obtained in each test are summarized for the proposed mixtures (Table 2). In general, the results between them were analyzed based on the reference mixture without any substitution (S0). The results were also compared with the works of Krishnamoorthy et al. [27]. The results presented in this paper had a higher percentage of substitution and a higher W/C ratio in addition to introducing the use of LFS slag. However, the similar evolution of the parameters studied for GGBS slag was checked.

### 5.1. Flow Cone Test Results

Flow cone test results are shown in Figure 3. It can be observed that there were no significant differences among the mixtures. The flow cone test results were always within an interval from 8.5 to 9 s. The main implication here is that the use of these types of slags as substitute of cement in cementitious grout had no significant effect in the fluidity of the resultant mix, at least in substitution percentages of up to 50%.

This is an important conclusion because these new mixtures will not present disadvantages in their application with respect to the original ones while being able to take advantage of the same implementation technologies.

### 5.2. Flexural Strength Test Results

Flexural strength test results are shown in Table 3 for 10 mixtures at 7, 28, and 90 days. In Figure 4, the data are grouped according to time and percentage of substitution for each slag type. Figure 5 also allows for analyzing the behavior of each mixture with respect to the reference. Each result curve (MR) is non-dimensioned with respect to the corresponding value of S0 (MR_S0_).

None of the mixes with slag substitution achieved the flexural strength reference (S0) at 7 days. However, at 28 days, the MR difference was reduced between S0 and two of the slag types (S1 and S2). One of the effects observed with slag was that the hardening process of the mixture was modified and delayed. As it has been observed in concrete mixtures incorporating GGBS slags as partial cement replacement, the strength at early stages was lower in comparison with traditional concrete. The results obtained in this work are in accordance with the results shown in previous works (Parron-Rubio et al. [7]).

No direct correlations were identified for each type of slag and its improvement with respect to MR.

S1 mixtures showed a higher flexural strength than S2 and S3 mixtures. In the literature, there are some works that indicate that pozzolanic materials with a high SiO_2_ content have better mechanical properties than pozzolanic materials with a low content of SiO_2_ [7,46]. This can be the reason why S1 mixtures showed the highest flexural strength due to their highest SiO_2_ content in comparison with S2 and S3 slags.

It seems that each slag had a particular dosage that optimized its behavior in the test. The S1_40 mix showed the best performance overall in this test, obtaining a flexural strength gain of up to 18.6% at 90 days with the 40% substitution.

S1 slags were unique in presenting an increase in flexural strength at 90 days for every substitution fraction (S1_30, S1_40, and S1_50). On the other hand, S3 slag showed poor results for this test. As it can be observed in Figure 4, flexural strength loss for this type of slag appeared at 7, 28, and 90 days. It seems that in this type of slag, the hardening stopped after 28 days. Actually, our conclusion is that from there it is maintained. We do not consider that the small decrease observed in Figure 4 is representative of any behavior, but is the consequence of some distortion of results.

### 5.3. Compressive Strength Test Results

Compressive strength test results are shown in Table 4 for 18 test specimens at 7, 28, and 90 days. In Figure 6, the data are grouped according to time and percentage of substitution for each slag type (S1, S2, and S3). Figure 7 also allows for analyzing the behavior of each mixture with respect to the reference. Each result curve (Rs) is non-dimensioned with respect to the corresponding value of S0 (Rss_0_).

As for the flexural strength results, none of the mixes obtained a compressive strength gain at 7 days with respect to the reference grout mixture. The behavior at 28 and 90 days of the different mixes differed depending on the type of slag and the slag–cement substitution percentage.

The compressive strength of S1 slag grew as the substitution percentage increased. The behavior of the 40% mixture was similar to the 30%. The best compressive performance is attributed to the 50% mixture, which obtained a strength gain at 28 and 90 days of 28.35% and 35%, respectively.

The behavior of S2 slag at 7 days was similar for every substitution percentage and was about 30% less than the reference grout mixture at the same age. S2 mix obtained strength loss at 28 days for every substitution percentage, with the 40% mix (S2_40) the one performing the best, followed by the 30% mix and 50% mix. However, the results for 90 days show a relationship between substitution percentage and compressive strength, with the latter being greater as the substitution percentage decreased.

The compressive strength loss obtained using L3 mixes, at every stage and percentage substitution, was significant. Furthermore, the loss was greater as the substitution percentage increased. As in the case of the flexural strength, the mixtures with GGBS slags showed a higher compressive strength than LFS slags.

Krishnamoorthy et al. [27] presents results of variation of compressive strength with GGBS slags. They test them from porous concrete blocks containing a mixture of supplementary cementitious materials including fly ash, GGBS, and silica fume. One of them was prepared with a grouting mixture of ordinary Portland cement, 40% GGBS as aggregate, water–cement ratio of 0.35, and 1% superplasticizer. Although it is a different product, we can observe a similar slight improvement in the compressive strength.

### 5.4. Exudation Test Results

The results of the exudation test are shown on the Figure 8.

It can be distinguished in the figure that the mixture without substitution of cement by slag was the one that showed a higher percentage of water exudation at 240 min, obtaining a value of almost 30%.

GGBS (S1) mixtures had low values of exudation at early stages up to 120 min and obtained the lowest percentages at 240 min, except for the S2 mixture with 50% substitution.

LFS1 (S2) slag showed an exudation behavior similar to the reference mixture, although they obtained water exudation percentages superior to those of the rest of the slags.

Regarding the percentage substitution, the results clearly show that with higher substitution percentages, the exudation decreased.

## 6. Conclusions

In this paper, experimental results obtained from grouts with cement substitutions by slags in a dosage of up to 50%, W/C ratio of 1, and no additives are presented. The results for different white slags are studied (GGBS and LFS). All the specimens have been tested for exudation, compressive strength, and flexural strength to analyze the feasibility of the mixture for industrial applications. According to the results described in previous sections, the following conclusions can be highlighted:

In general, the mixtures obtained show an improvement in factors such as fluency and viscosity. The slags had a lower density than cement and cause a mixture more fluid. This was an improvement in application where this factor was important.

The mechanical response was less homogeneous and depended greatly on the origin of the slag, as expected. Improvements were observed in the results of compression and bending strength for mixtures with S1 slags (10% in bending and 35% in compression test). However, slag types S2 and S3 gave rise to mixtures with losses of resistance of 85% in compression with respect to the reference. This was due to the lower content of SiO_2_ in slags S2 and S3 (LFS) in comparison with S1 slag (GGBS). It has been proven that there was a great difference between slags according to their origin, not only for its composition, but also for the treatment received prior to mixing. Therefore, each generator of slag waste required a study of the goodness of its product in terms of its use as a cementitious substitute. However, the tests seemed to indicate that an adequate treatment increased the potential valorizing of the waste in question.

In the conventional slurry, the aggregate was the least deformed; therefore, this was where the tensions were concentrated. They were then transferred to the ITZ. The green grouts (with cement substitution by slag) had a lower density rate. This increased the speed of the mechanism of the conservation equation of the mass. This filled the microstructural spaces, facilitating the adherence between the aggregate and the cement paste inside the transition.

In addition, the resultant cement grout was a sustainable material with a lower cost in comparison with traditional cement grouts.

The fundamental conclusion of this study was to verify the feasibility of obtaining mixtures with cement substitution by slag. A 50% reduction in the cement used in the mix was achieved, and at the same time, the viability of the grout was maintained. The fluidity obtained allows for use in applications where there can be an intensive use of this grouts, such as jet-grouting for mixtures S1 and S2, or ground improvements for S3 mixtures

Higher cement substitution levels using slag waste may also be possible but this would require further investigation.

GGBS slags improved the mechanical and workability capabilities of the resulting mixture. The LFS slags studied in this work can be employed in other types of works where a high strength is not required. Therefore, this conclusion presents an opportunity to improve waste slag valorization if we progress in the high-level percentage of substitution with slags in ordinary products such as grouts or concrete.

## Figures and Tables

**Figure 1 materials-12-01166-f001:**
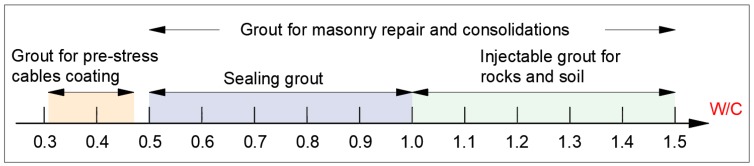
Schematic water–cement ratio (W/C) depending on the application.

**Figure 2 materials-12-01166-f002:**
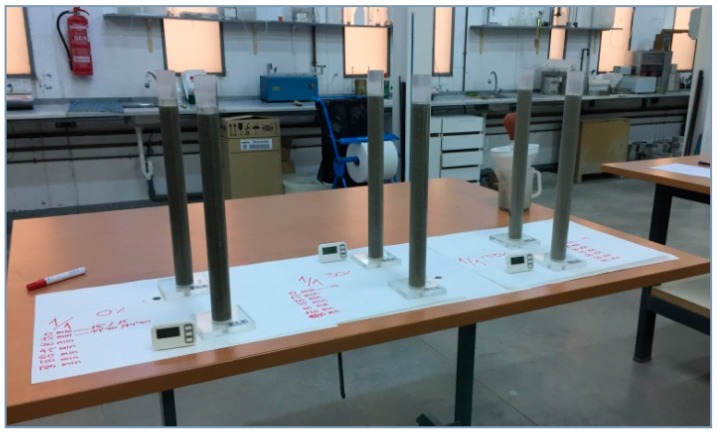
Exudation tests.

**Figure 3 materials-12-01166-f003:**
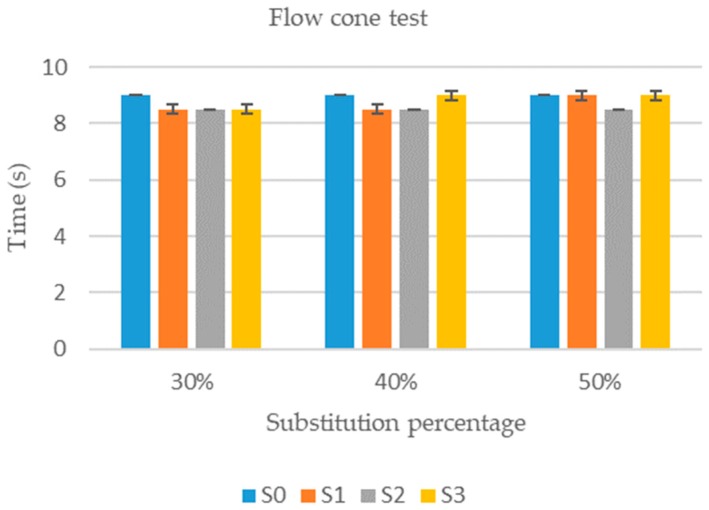
Flow cone test results.

**Figure 4 materials-12-01166-f004:**
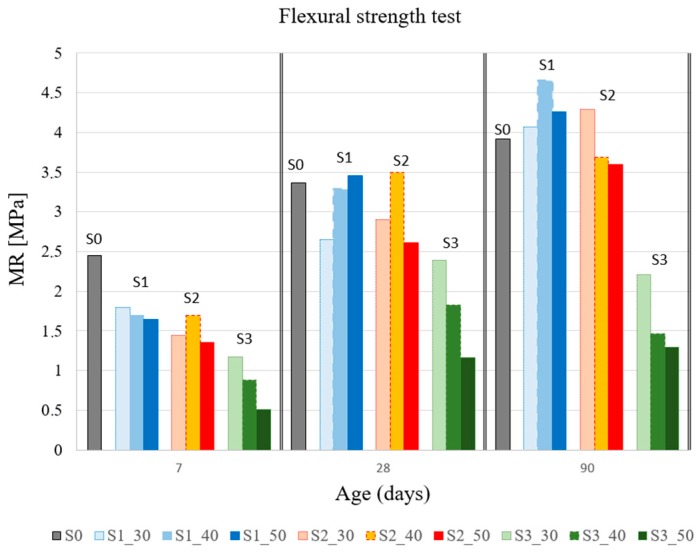
Results of flexural strength test (MR)(MPa).

**Figure 5 materials-12-01166-f005:**
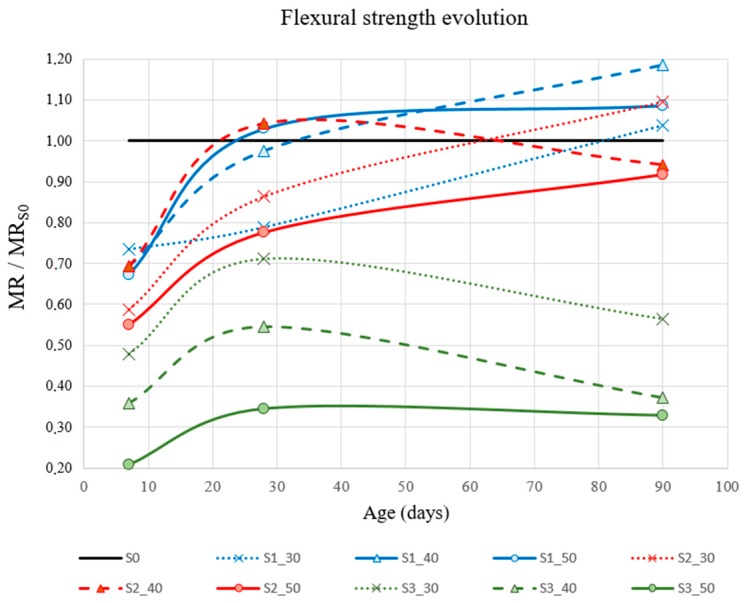
Flexural strength evolution (MR/MR_S0_).

**Figure 6 materials-12-01166-f006:**
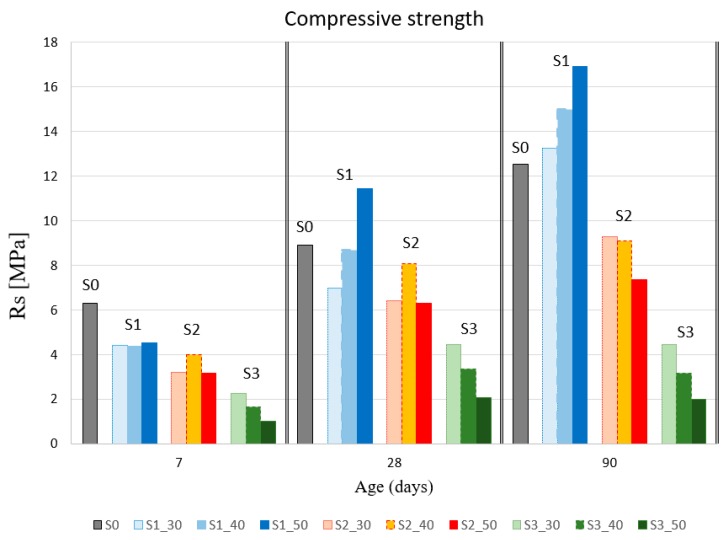
Results of compressive strength test (Rs)(MPa).

**Figure 7 materials-12-01166-f007:**
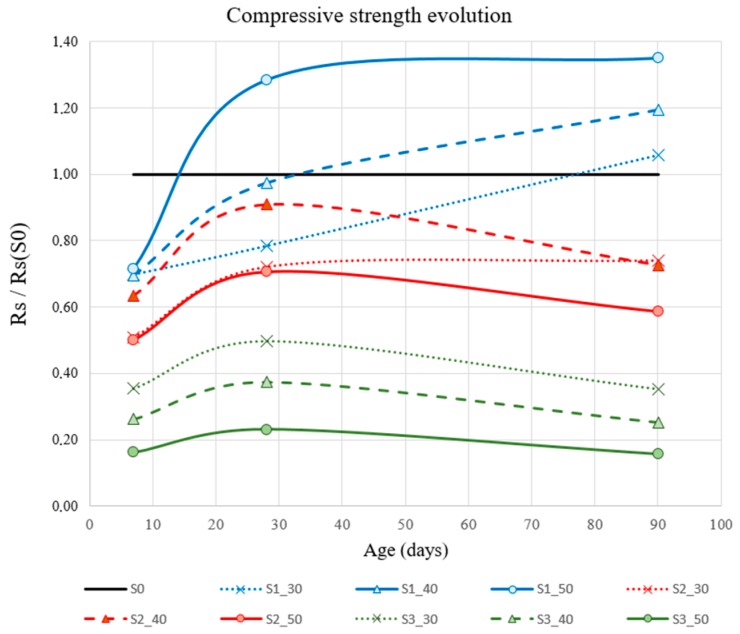
Compressive strength evolution (Rs/Rs(_S0_)).

**Figure 8 materials-12-01166-f008:**
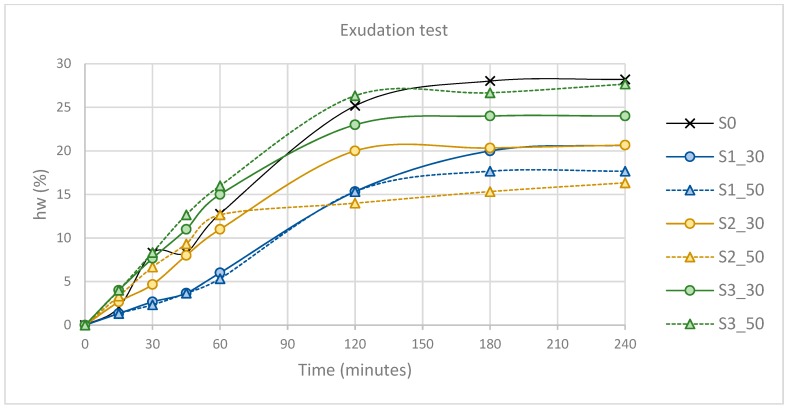
Exudation test.

**Table 1 materials-12-01166-t001:** Cement and slag chemical composition (data provided by the supplying company).

Slag Origin/Chemical Composition	SiO_2_	Al_2_O_3_	CaO	Fe_2_O_3_	MgO	Na_2_O	K_2_O
	%	%	%	%	%	%	%
Cement	20–22	4–10	55.62	4	2	0.3	0.3
Slag GGBS (S1)	35.9	11.2	40	0.3	7.7	0.2	0.4
Slag LFS 1 (S2)	22.28	9.37	56.94	0.84	7.37	0	-
Slag LFS 2 (S3)	15.85	16.53	57.56	0.83	7.7	-	-

**Table 2 materials-12-01166-t002:** Cementitious grout mixtures composition.

Mix Denomination	Slag	Substitution
S0	-	0%
S1_30	GGBS	30%
S1_40	GGBS	40%
S1_50	GGBS	50%
S2_30	LFS 1	30%
S2_40	LFS 1	40%
S2_50	LFS 1	50%
S3_30	LFS 2	30%
S3_40	LFS 2	40%
S3_50	LFS 2	50%

**Table 3 materials-12-01166-t003:** Flexural strength results.

Mixes	7	28	90	% Strength Gain at 90 Days
S0	2.45	3.36	3.92	-
S1_30	1.80	2.65	4.07	3.8%
S1_40	1.70	3.28	4.28	9.18%
S1_50	1.65	3.46	4.63	18.1%
S2_30	1.44	2.90	4.29	9.4%
S2_40	1.70	3.5	3.69	−5.9%
S2_50	1.35	2.61	3.60	−8.2%
S3_30	1.17	2.39	2.21	−43.6%
S3_40	0.88	1.83	1.46	−62.8%
S3_50	0.51	1.16	1.29	−67.1%

**Table 4 materials-12-01166-t004:** Compressive strength results.

Mixes	7	28	90	% Strength Gain at 90 Days
S0	6.29	8.89	12.52	-
S1_30	4.40	6.98	13.25	5.8%
S1_40	4.38	8.66	14.97	19.6%
S1_50	4.50	11.41	16.90	35.0%
S2_30	3.20	6.41	9.27	−26.0%
S2_40	3.98	8.08	9.09	−27.4%
S2_50	3.15	6.27	7.34	−41.4%
S3_30	2.23	4.42	4.42	−64.7%
S3_40	1.65	3.33	3.17	−74.7%
S3_50	1.02	2.05	1.98	−84.2%

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
