# Peer review of "Study of the Suitability of Different Types of Slag and Its Influence on the Quality of Green Grouts Obtained by Partial Replacement of Cement"

_materials, 2019, doi:10.3390/ma12071166_

Round 1

Reviewer 1 Report

The research undertaken is well in line with current trends in the development of composite technologies on cementitious binders, waste utilization and environmental protection. The article presents very basic, preliminary research done in a narrow scope. The authors emphasize that the article contains only part of the research done and the research is preliminary. The reviewer believes that in an article in a world-wide journal, research results should be published that have been made in a much wider scope and better documented. In this context, the reviewer considers the publication premature, and because the subject matter is interesting, he encourages the authors to expand it and only then to submit for publication. At the same time, attention should be paid to the following issues:·         The issues raised by the authors have already been the subject of numerous studies. This should be better taken into account in the literature section (introduction). The authors should refer to previous research in the field of applying steel production waste slags, the achievements in this field are wide. What is new to the state of knowledge is brought by the conducted research. ·         It is not necessary to discuss in detail the typical methods used. ·         For what purpose is included in the article 4.1 chapter? ·         How do the authors explain the loss of strength after 90 days of S3 grouts? ·         What are the practical implications and applicability of slags to injection slurries due to their technical properties? ·         The discussion of the obtained results should be extended, referring the results to those known from the literature. ·         The conclusions in lines 357 and 360 do not seem to result from the research and are speculative. ·         The conclusion in row 366 is an indication for further research, with the authors should be aware that the research of steel manufacturing waste slags has already been carried out and in a wide range.

Author Response

Response to Reviewer 1 Comments

Point 1: The research undertaken is well in line with current trends in the development of composite technologies on cementitious binders, waste utilization and environmental protection. The article presents very basic, preliminary research done in a narrow scope. The authors emphasize that the article contains only part of the research done and the research is preliminary. The reviewer believes that in an article in a worldwide journal, research results should be published that have been made in a much wider scope and better documented.

Response 1: The literature and the introduction have been corrected. This paper presents results based on an original methodology little comparable with other authors. For example, the water-cement ratio used or the one that does not use any type of additive, as superplasticiser agents. It also does not seek to improve any characteristics of existing slurries, but rather we seek a product where we can use steel slag intensively, guaranteeing minimum resistance and workability.

However, we appreciate the advice to improve this work, so the literature has been updated focussing on recent contributions of authors related to cementitious grouts containing supplementary cementitious materials.

On the other hand, the introduction has been rewritten not to cause confusion about the research presented. It is not a preliminary research but a first phase of study where we analyze the most appropriate slag typology with a view to further increasing the percentage of substitution in subsequent experimental campaigns.

Point 2  The issues raised by the authors have already been the subject of numerous studies. This should be better taken into account in the literature section

Response 2: As it is mentioned before, the literature has been updated

Point 3 The authors should refer to previous research in the field of applying steel production waste slags, the achievements in this field are wide. What is new to the state of knowledge is brought by the conducted research.

Response 3:  We present a grout manufacturing methodology where the main objective is to achieve a mixture with the properties of a cementitous grout with the minimum percentage of cement in it. The point of view is not to improve the mixture for a specific problem but to be able to valorize as much slag as possible in usual uses. However, in previous works with concrete, important variations were observed depending on the slag used. So we have to establish how the origin of it and its previous treatment make it appropriate or not. It is a point of view focused on the environmental viability of a new product. Therefore, we introduce higher percentages of substitution than other works, with higher water to cement relations too, and no chemicals additives.

Point 4: It is not necessary to discuss in detail the typical methods used.

Response 4:  Some of the figures (figures 2 and 4) are retired and some paragraphs have been rewrite to reduce this section.

Point 5:  For what purpose is included in the article 4.1 chapter?

Response 5:  This section has been removed from the results and conclusions and relocated. Indeed, it is not a contribution (results) of this investigation. We thought it was interesting to remember the treatment that this theory gives to these mixtures. Especially the hypothesis that some characteristics of the mixture after the substitution undergo little modification, for example, the density.

Point 6: How do the authors explain the loss of strength after 90 days of S3 grouts?

Response 6:  Our experience is that for this type of slag the hardening stops after 28 days. Actually, our opinion is that from there it is maintained. We do not consider that the small decrease observed in the figure is representative of any behavior, but the result of some distortion of results due to some test specimen in bad condition.

Point 7: What are the practical implications and applicability of slags to injection slurries due to their technical properties?

Response 7:  For our purposes, the main characteristic is that we can achieve a 50% reduction in the cement used in the mix and, at the same time, maintain the viability of the grout. With the water / cement ratio used, we think of high volume injection applications. Jet-grouting for mixtures S1 and S2, or ground improvements for S3 mixtures that have a lower mechanical strength.

Point 8: The discussion of the obtained results should be extended, referring the results to those known from the literature.

Response 8:  The discussion has been updated with some results as the Krishnamoorthy et al. ones.

Point 9: The conclusions in lines 357 and 360 do not seem to result from the research and are speculative.

Response 9:  this conclusion has been removed.

Point 10: The conclusion in row 366 is an indication for further research, with the authors should be aware that the research of steel manufacturing waste slags has already been carried out and in a wide range.

Response 10:  this conclusion has been rewrite. However, we think that this research presents an opportunity to improve waste slag valuation if we progress in high-level percentage of substitution with slags in ordinary products as grouts or concrete.

Reviewer 2 Report

The author's work is s suitability of different slag (GGBS and 1 LFS) and its influence on the quality of green grouts obtained by partial replacement of cement. However, in order to improve the completeness of the paper, I think there are some points to be supplemented below.

 - The main contents that I would like to emphasize in this paper need to make sure that they point out the usage rate of the material itself or point out the possibility of using it as grout material.

 - Additional explanations are needed for the performance requirements of grout material.

 - The author is considered to focus on the part related to the test compared to the description of the materials used. Please give a detailed explanation of LFS, a relatively unfamiliar material.

 - There is a lack of analysis of the experimental results presented by the author. Use the citations and add the author's argument about the experimental results.

 - Grout material expansion and contraction, the mechanism that acts from the ground, please present if there is data such as stability.

 - Please review the title of the article according to your research content.

Author Response

Response to Reviewer 2 Comments

Point 1: The topic is of general interest, and the presentation is relatively clear.

This article contains very interesting new aspects, but in manuscript the authors must underline the major findings of their work and explain how the use of their proposed procedures and resulted materials represents a progress to other similar published papers. The novelty must be pointed.

Response 1: This paper presents results based on an original methodology little comparable with other authors. For example, the water-cement ratio used or the one that does not use any type of additive, as superplasticiser agents. It also does not seek to improve any characteristics of existing slurries, but rather we seek a product where we can use steel slag intensively, guaranteeing minimum resistance and workability.  However, we appreciate the advice to improve this work, so the introduction has been updated.

Point 2: The Abstract must be rewrite, some experimental data are necessary, not only 50% replacement.

Response 2: The abstract has been rewrite to be better introduce the work presented.

Point 3: Material must be improved. The u.m for properties must be in SI.

Response 3: The materials units has been improved to S.I.

Point 4: Table 2 must be modified LFS1 (in table 1) and LFS from location 1. The column with CEM I 42,5 is not necessary, and W/C is same for all mixtures.

Response 4: The tables have been modified.

Point 5: Please clarified: in conclusion the authors present “The results for different white slags are studied”, in Introduction “The EAF slags can be also divided in two types: oxidizing or black and reducing or white”, but in materials are presented mixtures with LSF and GGBFS.

Response 5: The paragraph has been rewrite. GGBS are Blast Furnace Slag. The EAF are Electric Arc Furnace. The EAF slags can be also divided in two types: oxidizing or black (EAFS) and reducing or white (LFS).

Point 6: Tests description must be rewrite. Figures 2, and 4 aren’t necessary. The author presents EN standard numbers and types of devices. This chapter together 4.1 are too long.

Response 6: Figures (figures 2 and 4) are retired and some paragraphs are rewritten to reduce this section.

Point 7: 4.1. Interfacial Transition Zone Review… is not result and discussion!

Response 7: The chapter 4.1 has been removed from the results and conclusions and relocated.

Point 8: In text is too much we: challenges we face as a society is to achieve a balance between …., … environment where we can satisfy our needs without…, … aware that we inhabit a world with finite resources… etc. In my opinion in the technical literature, impersonal write is preferred.

Response 8: The writing style has been modified as the review suggests.

Point 9: MR? please explain… At 273 each result curve (MR) ….is not clear.

Response 9: The flexural strength is expressed as MR (modulus of rupture) abbreviation.

Point 10: In Materials: prismatic test specimens of 4 × 4 × 16 cm were prepared. Please verify if for compressive strength the standard recommend prisms or cubes.

Response 10: effectively the standard test specimens for compressive strength are cubic. In the laboratory it has been used the resulting specimen, after flexural strength tests. This fragment of the prismatic specimen is used in the compression test. According to Neville [43] the compressive strength of the modified cubic specimen would be a 5% higher than the standard cubic specimen.

Point 11: The presentation reflects the present state of knowledge.

Point 12: The references from last years are present, but references [9] wasn’t cited.

Response 12: The reference [9] has been added to the text.

Point 13: The text is easy to understand by scientists in other disciplines.

Point 14: The guide of authors is respected.

Point 15: The Conclusion must be rewrite. 

Response 15: The conclusion has been updated.

Point 16: Please present comparison with other studies about same subject.

Response 16: The discussion has been updated with some results as the Krishnamoorthy et al. ones.

Point 17: The English must be revised.

Response 17: The text has been revised with minor changes.

Reviewer 3 Report

The paper “Study of the Suitability of Different Slag (GGBS and LFS) and its Influence on the Quality of Green Grouts Obtained by Partial Replacement of Cement”, authors Francisca Perez-Garcia, Maria Eugenia Parron-Rubio, Jose Manuel Garcia-Manrique and  Maria Dolores Rubio-Cintas

The manuscript Can be published in Materials after MAJOR REVISION

In this paper is presented the work methodologies for the manufacture of green  cementitious grout mixture for cement replacement.

Observation:

The topic is of general interest, and the presentation is relatively clear.

This article contains very interesting new aspects, but in manuscript the authors must underline the major findings of their work and explain how the use of their proposed procedures and resulted materials represents a progress to other similar published papers. The novelty must be pointed.

The Abstract must be rewrite, some experimental data are necessary, not only 50% replacement.

The key words permit found article in the current registers or indexes

Material must be improved. The u.m for properties must be in SI. Table 2 must be modified LFS1 (in table 1) and LFS from location 1. The column with CEM I 42,5 is not necessary, and W/C is same for all mixtures.

Please clarified: in conclusion the authors present “The results for different white slags are studied”, in Introduction “The EAF slags can be also divided in two types: oxidizing or black and reducing or white”, but in materials are presented mixtures with LSF and GGBFS.

Tests description must be rewrite. Figures 2, and 4 aren’t necessary. The author presents EN standard numbers and types of devices. This chapter together 4.1 are too long.

4.1. Interfacial Transition Zone Review… is not result and discussion!

In text is too much we: challenges we face as a society is to achieve a balance between …., … environment where we can satisfy our needs without…, …aware that we inhabit a world with finite resources… etc. In my opinion in the technical literature, impersonal write is preferred.

MR? please explain… At 273 each result curve (MR) ….is not clear.

In Materials: prismatic test specimens of 4 × 4 × 16 cm were prepared. Please verify if for compressive strength the standard recommend prisms or cubes.

The presentation reflects the present state of knowledge.

The references from last years are present, but references [9] wasn’t cited.

The text is easy to understand by scientists in other disciplines.

The guide of authors is respected.

The Conclusion must be rewrite.  

Please present comparison with other studies about same subject.

The English must be revised.

MAJOR REVISION

Author Response

Response to Reviewer 3 Comments 

The author's work is s suitability of different slag (GGBS and 1 LFS) and its influence on the quality of green grouts obtained by partial replacement of cement. However, in order to improve the completeness of the paper, I think there are some points to be supplemented below.

Point 1: The main contents that I would like to emphasize in this paper need to make sure that they point out the usage rate of the material itself or point out the possibility of using it as grout material.

Response 1: The introduction has been extended to emphasize the main objective of the study: to achieve the use of slurries where the use of slag waste is intensive. With the minimum cement possible. This work analyzes different origins of the slag to study the viability of each one.

Point 2: Additional explanations are needed for the performance requirements of grout material.

Response 2: The desirable properties for the grouts are that they must possess good fluidity, reduction of bleeding, initial setting time not too short, adequate strength and durability. The result discussion has been updated to contrast the results with these properties.  

Point 3: The author is considered to focus on the part related to the test compared to the description of the materials used. Please give a detailed explanation of LFS, a relatively unfamiliar material.

Response 3: The introduction has been expanded to better describe this material. In summary, they are slag from the refining process in the production of steel by electric arc furnace.

Point 4: There is a lack of analysis of the experimental results presented by the author. Use the citations and add the author's argument about the experimental results.

Response 4: The result section has been updated to expand the analysis and to compare results with other authors.

Point 5: Grout material expansion and contraction, the mechanism that acts from the ground, please present if there is data such as stability.

Response 5: They are indeed interesting properties for these materials. Unfortunately, we do not currently have data on this. However, we see very interesting his analysis in future research.

Point 6: Please review the title of the article according to your research content.

Response 6: In our opinion, the title is appropriate. As mentioned before the introduction and results have been rewrite to introduce more dates of the suitability of this product to use as grouts in common uses. However, if the reviewer has any suggestion it will be very appreciated.

Round 2

Reviewer 1 Report

I accepting corrections made. 

Reviewer 2 Report

I look forward to interesting research on related fields in the future. Thank you for your response.